# Gestational and Lactation Exposure to Perfluorohexanoic Acid Results in Sex-Specific Changes in the Cerebellum in Mice

**DOI:** 10.3390/ijms26168008

**Published:** 2025-08-19

**Authors:** Elizabeth C. Plunk, Navya Navnith, Hannah Swan, Linh Le, Matthew McCall, Marissa E. Sobolewski, Ania K. Majewska

**Affiliations:** 1Department of Environmental Medicine, University of Rochester Medical School, Rochester, NY 14642, USA; elizabeth_plunk@urmc.rochester.edu (E.C.P.); marissa_sobolewski@urmc.rochester.edu (M.E.S.); 2Environmental Health Science Center, University of Rochester Medical School, Rochester, NY 14642, USA; 3Department of Neuroscience, University of Rochester Medical School, Rochester, NY 14642, USA; navyanavnith@gmail.com (N.N.); linhle0514@gmail.com (L.L.); 4Department of Biostatistics and Computational Biology, School of Medicine and Dentistry, University of Rochester, Rochester, NY 14642, USA; hannah_swan@urmc.rochester.edu (H.S.); matthew_mccall@urmc.rochester.edu (M.M.); 5Center for Visual Science, University of Rochester, Rochester, NY 14642, USA

**Keywords:** per- and polyfluoroalkyl substances, bulk-RNA-sequencing, Purkinje cells, microglia

## Abstract

Currently regulated per- and polyfluoroalkyl substances (PFAS) have been associated with immune, endocrine, and neurotoxicity following gestational exposures. As a result, industries have effectively replaced them with next-generation PFAS, including perfluorohexanoic acid (PFHxA). PFHxA is increasingly found in the serum of pregnant women and in breast milk, and adult human post-mortem studies indicate that PFHxA is found in the brain, with the highest concentrations in the cerebellum and hypothalamus. Despite evidence of gestational, lactational, and nervous system exposure to PFHxA, developmental neurotoxicity (DNT) testing in mammals has not been conducted. For DNT evaluation, we exposed pregnant C57Bl/6J mice daily from gestational day 0 through postnatal day (P) 21 to two PFHxA exposure levels (a lower (0.32 mg/kg of body weight (bw), or higher (50 mg/kg of bw) dose of PFHxA)) or ddH_2_O using treat-based administration. Given the high PFHxA levels in the cerebellum in post-mortem studies and the cerebellum’s protracted developmental window, we assessed acute transcriptional dysregulation and cellular morphology in this brain region on the last day of exposure at P21. Using bulk-RNA sequencing, we found that PFHxA exposure had subtle effects on transcripts related to neurons and glia, with females having a greater number of dysregulated transcripts than males. Using immunohistochemistry, we found that Purkinje cell linear frequency was increased in specific lobules in the higher-exposure group and that microglial morphology underwent subtle changes in specific cerebellar layers in the lower-exposure group in both sexes. Together these data suggest that PFHxA exposure may have lobule-specific impacts on the development of both neurons and glia in the cerebellum, highlighting the importance of studying the neurotoxicity of PFHxA in both sexes.

## 1. Introduction

Per- and polyfluoroalkyl substances (PFAS) have been used in consumer products since the 1950s due to their stain-proofing and water-repelling properties. PFAS are made up of a carbon chain backbone with at least one of the carbons being bound to a fluorine [1]. This polar covalent carbon-to-fluorine bond is one of the strongest organic chemical bonds to occur in nature and confers the stability that prevents PFAS from being easily degraded in the environment [2]. Additionally, with a hydrophobic carbon backbone and a hydrophilic functional group, PFAS are mobile in both aqueous and lipophilic environments such as water [3] and soils [4], respectively, resulting in the contamination of the global food supply. Wildlife and humans have been exposed to PFAS for decades since their introduction, and exposure will continue as these stable compounds remain in the environment [5]. In fact, it is predicted that over 97% of Americans have detectable blood levels of PFAS [6].

The large scale of human exposure throughout the lifespan has led researchers to focus on understanding the possible health effects of PFAS. According to the Environmental Protection Agency (EPA) of the United States of America, 14,000 different PFAS chemicals have been detected in the environment [7], and even the PFAS that have been phased out of production (often referred to as “legacy PFAS”) are still lingering [8]. In fact, the majority of research has focused on understanding the effects of these legacy PFAS on human health. However, recent studies have shown that despite the large number of different PFAS that have been produced, the majority detected in the environment are the “new” PFAS [9], whose effects on human health have not been carefully investigated. Investigation of the newer PFAS is especially necessary in light of the fact that several legacy PFAS, like perfluorooctanoic acid (PFOA) and perfluorooctane sulfonic acid (PFOS), were phased out of production after studies showed their detrimental human health effects [10].

Neurodevelopment might be a particularly vulnerable period as developmental PFAS exposures correlate with broad behavioral effects in humans including in motor [11,12], language [12], and attention/activity [13] domains. Additionally, each individual PFAS, as well as each distinct combination of PFAS, has the possibility of eliciting unique effects. It is unclear, however, which property of individual PFAS confers toxicity. Some scientists have postulated that long-chain PFAS, defined as those containing at least a 7-carbon chain backbone, are more toxic than short-chain PFAS, while others have suggested that it is the different functional groups, irrespective of carbon chain length, that confer toxicity [1]. Multiple mechanistic targets of PFAS toxicity have been identified. It is interesting to note that one group found that exposure to the same dose of PFOS and PFOA, which have the same number of carbons but different functional groups, had different effects on neurons and microglia in zebrafish [14], suggesting that the different functional groups of these two PFAS determine their mechanism of action.

One common new PFAS that has not received broad DNT testing is perfluorohexanoic acid (PFHxA). PFHxA has a short-chain, six-carbon backbone with a carboxylic acid tail group. While PFHxA has been described as less potent for renal toxicity than PFOA [15], it unclear whether its toxicity to other organs is different than that of legacy PFAS. Considering that PFHxA has been found in higher concentrations in the human brain and in the liver compared to other PFAS, including PFOS and PFOA [16], studies investigating exposures to PFHxA throughout neurodevelopment are critical to understanding the toxicity of PFHxA.

We have previously reported that PFHxA exposure during gestation and lactation has sex-dependent behavioral effects in adult offspring [17]. Here, we assayed whether this same exposure affects gene expression and cellular profiles in the cerebellum, alterations in which could underlie long-term behavioral changes. Our evaluations began with the cerebellum due to its protracted development [18], which could make it more vulnerable to this developmental toxicological exposure. Furthermore, the cerebellum was one the brain regions with the highest reported PFHxA concentrations in humans [19]. Importantly, while the cerebellum is traditionally recognized for its role in motor functions, over the past decade its role in cognitive domains has been appreciated [20], suggesting that disruptions to cerebellar development could have profound repercussions for behavior. In fact, the cerebellum has been implicated in anxiety-like behaviors [21] and social behavioral domains [22], and perturbation of cerebellar cells and circuits has also been investigated in neurodevelopmental disorders like autism spectrum disorder [23,24], attention deficit hyperactivity disorder [25], and fetal alcohol spectrum disorder [26]. We hypothesized that developmental PFHxA exposure disrupts the development of neuronal and glial cerebellar populations. We tested this hypothesis by using RNA sequencing of the cerebellum at the time of weaning, coupled with immunohistochemical analyses probing Purkinje cells, astrocytes, microglia, and myelin at the same time point. Our study sheds light on how the different cell types of the cerebellum are affected by developmental PFHxA exposure and could provide insights into cellular substrates that contribute to PFHxA-dependent changes in brain function.

## 2. Results

To determine how gestational and lactational exposure to PFHxA affects cerebellar development, pregnant C57Bl/6 mice were exposed daily from gestational day 0 to the time of weaning (P21 for the pups) to either vehicle or two doses of PFHxA (lower: 0.32 mg/kg of body weight (bw); higher: 50 mg/kg of bw; Figure 1A). On P21, the whole cerebellum from one female and one male pup per litter was harvested and subjected to either bulk-RNA sequencing or immunohistochemistry to determine transcriptomic or cellular profiles, respectively, that are sensitive to PFHxA exposure (Figure 1B). Bulk-RNA sequencing revealed that overall, PFHxA exposure resulted in only subtle cerebellar transcriptomic changes on P21, as observed in the principal component analysis (PCA) plots, where both females (Figure 1C) and males (Figure 1D) from each treatment group largely overlap, and the variance captured in the top two principal components is modest. To better understand any subtle changes in the cerebellar transcriptome elicited by PFHxA, we focused on the lower-exposure group first and analyzed significantly dysregulated genes in PFHxA-exposed males and females compared to their controls (Figure 2). Volcano plots showed significantly more up- and downregulated genes in females (2328 genes) than males (949 genes; Figure 2A,B), with the majority of genes being downregulated by PFHxA exposure. Of the 1944 genes downregulated by PFHxA exposure in females, only 88 genes were also downregulated in males (Figure 2C). GO biological process analysis revealed that the top five downregulated pathways did not overlap between the sexes, with developmental pathways being downregulated in females (Figure 2E), and pathways related to RNA biology being downregulated in males (Figure 2G). Similar to downregulated genes, there were only 87 genes upregulated in both males and females by low-dose PFHxA exposure with a larger number of sex-specific upregulated genes (1395 in females vs. 373 in males; Figure 2D). In females, the top five upregulated pathways were related to RNA splicing (Figure 2F). In males, the top five upregulated pathways included regulation of cell morphogenesis and protein regulation (Figure 2H). Together, these data suggest that lower PFHxA exposure affects neurodevelopmental pathways in males and females differently.

When comparing the higher-exposure group to controls, we again found fewer upregulated than downregulated genes in both sexes, with females having a larger number of dysregulated genes than males (Figure 3A,B). Interestingly, females showed fewer dysregulated genes than in the comparison between the lower-exposure group and controls (1675 vs. 3426), while males had more dysregulated genes (1246 vs. 947). Of the 1011 genes downregulated in females, only 52 were also downregulated in males (Figure 3C). The top five downregulated pathways in females were related to synapse and dendrite development (Figure 3E), while the top five downregulated pathways in males included pathways involved in endothelial cell development and structure development (Figure 3G). Similar to the findings in the downregulated genes, females also had more upregulated genes (664) than males (432), with only 24 shared between the two sexes (Figure 3E). The pathways most upregulated in females included cellular development and regulation (Figure 3F). In males, the pathways most upregulated included protein localization in the synapse (Figure 3H). These data reveal that, similar to lower exposure, higher PFHxA exposure results in different pathway dysregulation between females and males.

Because lower-dose exposure resulted in the regulation of twice the number of genes than higher-dose exposure in females but not males, we wanted to understand whether these two exposure concentrations regulated a similar gene set by comparing the dysregulated genes in the lower and higher groups. Despite high numbers of genes being downregulated in females by lower-dose exposure compared to control (1483 genes; Figure 2A), only 403 (21%) of these were also in the downregulated gene set (1012 genes; Figure 3A) in females when the higher-exposure group was compared to control (Appendix A). Similarly, for upregulated genes in females, only 316 (21%) were downregulated in common in the comparisons between lower exposure and controls and between higher exposure and controls (Appendix A). A similar distinction between genes regulated by the two doses was found in males, with only 83 (17%) downregulated genes and 118 (26%) upregulated genes being shared by the two exposure groups (Appendix A). To compare these exposure groups directly, we redid our analyses comparing lower- and higher-exposure groups in males and females, suspecting that these would show large differences between the transcriptomic landscapes of the two exposure groups. In fact, we found a large number of significantly dysregulated genes between the two exposures that resembled comparisons of the lower-exposure group to control in both females (2184 vs. 2328) and males (1605 vs. 949) (Appendix A). Of the 1364 genes downregulated in females, only 178 genes were downregulated in males (Appendix A). Females also had more upregulated genes than males, with only 75 overlapping genes between the sexes (Appendix A).

Overall, this shows that the majority of genes dysregulated by gestational and lactational PFHxA exposure are dose- and sex-dependent, suggesting that low-dose exposure has a distinct and more profound effect on transcription than higher-dose exposure, especially in females. Interestingly, while both sexes showed dysregulated pathways, the lack of overlap between these pathways suggests sex-dependent effects of both exposures. This differential effect may indicate that this lower exposure concentration may have altered cell number, influencing differential gene expression. To address this, we evaluated cell-type-specific transcriptional profiles and morphology. We decided to use the female gene sets, which were larger than those for the males, to explore possible cellular-specific pathways that may be a target for PFHxA using curated cerebellar gene lists and immunohistochemistry for specific cell types.

### 2.1. Purkinje Cells

We first address Purkinje cells (PCs), the main output neuron of the cerebellum. In females, 134 genes in lower exposure, 26 in higher exposure, as well as 20 genes that were common to both exposures were dysregulated in a gene set that represented PC-related genes (Figure 4) [27]. Along with PC-related genes, genes involved in both neurotransmitter transport and axon guidance were dysregulated, suggesting that cerebellar neurons may be sensitive to PFHxA exposure (Appendix A). Moreover, apoptosis- and phagocytosis-related genes, which represent pathways that allow for the developmental pruning of excess PCs, were dysregulated (Appendix A), which could indicate perturbations in PC monolayer formation. While bulk-RNA sequencing data can reflect changes in cell type abundance, the number of both significantly up- and downregulated genes made it difficult to ascertain whether cell number or cellular function was being affected by these exposures. The dysregulated genes highlight that multiple processes are possibly targeted by PFHxA. For example, Lrrc24 and Ptgfrn are both expressed throughout the lifetime by PCs, but their biochemical function is unknown, while Grfa2 and Itga3 are involved in PC development [28,29] and are downregulated and upregulated, respectively, and Cttnbp2 is a synaptic gene that is downregulated and has been implicated in neurodevelopmental disorders [30,31]. These broad changes could indicate multiple functional changes within PCs.

To address changes in cell numbers, we utilized immunohistochemistry for calbindin, a reliable PC marker [32], to quantify PCs. We found that, overall, only females in the higher-exposure group had a significant increase in PC count in the whole cerebellum (Figure 4C,E). We then calculated the PC linear frequency, to normalize for cerebellum size, of each lobule. Interestingly, in both males and females, we found a significant increase in PC linear frequency in the higher-treatment group in only lobule III (Figure 4D,F, Appendix A).

### 2.2. PFHxA Effects on Glial Populations

First, we sought to understand if oligodendrocyte lineage cell-related genes were dysregulated. To do this, we curated a gene list that contained genes expressed by oligodendrocytes and their precursor cells [27]. Analysis of females revealed 45 dysregulated genes related to oligodendrocyte precursor cells (OPCs) (Figure 5A), and 235 dysregulated genes related to oligodendrocytes (Figure 5B). Genes related to OPCs and oligodendrocytes were largely downregulated in mice in the lower-exposure group, and this effect was more pronounced for oligodendrocytes than OPCs. To determine whether downregulation of OPC and oligodendrocyte genes resulted in altered myelination, we analyzed the percent area covered by myelin basic protein (MBP) immunoreactivity in the granule cell layer of each lobule in the cerebellum. There were no changes in males or females in either treatment group of MBP immunoreactivity in any lobule (Figure 5D,E, Appendix A), suggesting that at this time point, transcriptomic changes do not reflect changes in gross myelination in this layer of the cerebellum.

Myelination is not the only glia-related process that is a target of toxicant exposures. Astrocytes have also been reported to be affected by toxicant exposures, specifically PFAS exposures [33,34]. Following PFHxA exposure in females, 120 astrocyte-related genes were significantly downregulated in the lower-exposure group compared to controls, but this was not seen in the higher-exposure group, where only 11 astrocyte-related genes were altered (Figure 6A). As this downregulation could be due to a decrease in the number of astrocytes, we analyzed the percent area covered of GFAP in lobule IV/V of the cerebellum. Since GFAP labels Bergmann glia, which are in the Purkinje cell and molecular layer (PC and ML) [35], astrocyte cell bodies, and primary processes in the granule cell layer (GCL) [36], we analyzed the layers separately. There were no changes in the percent area covered in males or females in either treatment group (Figure 6C–F). Similarly, there were no changes in GFAP percent area covered in males or females in either treatment group in the GCL or combined PC and ML in other cerebellar lobules (Appendix A) or in the hippocampus or corpus collosum (Appendix A).

As astrocytes are known to play important roles in neuroinflammation, we next examined neuroinflammation-regulated genes and found that they were similarly downregulated in the lower-exposure group in females compared to controls (Appendix A), with 57 genes regulated in the lower-exposure and 10 in the higher-exposure group. Immune-related genes are expressed by many cell types, and many cell types could be involved in neuroinflammatory pathways. However, a recent study carried out in zebrafish suggested that immune cells may be vulnerable to PFHxA exposure during development [37], and a previous study in zebrafish suggested that microglia are vulnerable to developmental PFOS exposure [14]. Several microglia-related genes, Cx3cr1, P2Y12, and C1qc, were downregulated in the lower-exposure group compared to controls in females (Figure 7A, Appendix A), suggesting a possible loss of microglia or change in microglial function (related to homeostatic conditions or the complement system) after lower-dose exposure in females. Considering all of this, we decided to focus on microglia, the resident immune cells of the brain. We analyzed microglia density in the GCL and combined PC and ML of lobule IV/V of the cerebellum (Figure 7C–F) and the CA1 and dentate gyrus (Appendix A) and found no changes in either sex in either of the treatment groups.

While this suggests that PFHxA exposure does not directly impact microglial numbers, the transcriptomic analysis points to possible changes in the function of PFHxA-exposed microglia. We therefore decided to analyze microglia morphology using Sholl analysis, which can provide insights into changes in microglial structure that could reflect subtle changes in microglial function. Microglia in both cellular layers and in both sexes and treatments on average had ~3 processes emanating from the soma and peaked at ~5 processes at a distance of ~12 µm from the soma. We employed a nuanced analysis examining the effects of PFHxA exposures on microglia morphology. Using Bayesian Hierarchical Sholl models [38], we obtained 95% credible intervals that suggested that PFHxA exposure elicited changes in microglia morphology in females and males in a layer- and treatment-specific manner. In females, there were no changes in the GCL in either exposure group (Figure 8A,B), but we did see an effect of lower exposure on the critical value ϒ, which represents the width of the Sholl curve when compared to control in the PC and ML (Figure 8C,D), suggesting that the microglial arbor was larger in the PFHxA-exposed animals. Meanwhile, in males in the GCL, we saw an effect in lower exposure compared to controls, but not in higher exposure (Figure 8E,F), specifically in branch maximum τ, which represents the maximal number of intersections (or the height of the Sholl curve) increasing in PFHxA-exposed animals. In the PC and ML, there were no changes in either treatment group in males (Figure 8G,H). These data together indicate that microglia in different cellular layers are affected in females and males in the lower-dose exposure group.

## 3. Discussion

Overall, gestational and lactational exposure to PFHxA resulted in a dysregulation of transcripts in both males and females. However, these effects were subtle, as evidenced by the lack of separation between groups in the PCA and the fact that most genes did not exceed a 2-fold change. The subtle PFHxA-induced changes were more pronounced in females, with dysregulated transcripts related to Purkinje cells, immune cells, astrocytes, oligodendrocytes, and oligodendrocyte progenitor cells. All of the cell types and pathways analyzed were more perturbed in the lower-exposure group than the higher-exposure group, suggesting non-monotonic effects. Interestingly, there was little overlap in the dysregulated genes in males and females, suggesting sex-specific molecular or cellular targets of PFHxA in the developing brain. This leaves an exciting opportunity for future research to understand the sex differences of PFHxA neurotoxicity in mammalian models.

### 3.1. Neuronal Effects

The transcriptomic analysis of differentially expressed genes related to Purkinje cells suggests that multiple functions could be affected including maturation, plasticity, and synaptic function. Changes in PC numbers have garnered a lot of attention in cerebellar ataxias, where loss of PCs results in altered gait [39,40]. Interestingly, maternal immune activation during gestation has been shown to result in deficits in PCs in a lobule-specific manner in offspring in mice [41], mirroring the lobule-specific effects we observe with PFHxA. Early postnatal ethanol exposure has also been reported to result in decreased PC linear frequency [42], suggesting that PCs may be particularly sensitive to developmental insults. In fact, we have previously reported that gestational and lactational exposure to PFHxA results in subtle alterations in gait in adulthood [17]. However, the developmental effects currently described in the literature involve the loss rather than an increase in the number of PCs. The increased PC linear frequency that we report, specifically in lobule III, could be a reflection of delayed PC monolayer formation in a lobule-specific manner in the higher-treatment group. Additionally, lobule III is anteriorly located [43] and expands earlier than posteriorly located lobules, potentially making it more vulnerable to developmental insults [44].

### 3.2. Glial Effects

PFOA and PFOS exposures have been found to perturb glial cell dynamics, including effects on oligodendrocytes [45], astrocytes, and microglia [14]. Since these cell types have been reported to be vulnerable to PFAS exposures, it is not surprising that we found dysregulated genes related to all of these cell types at the transcriptional level, in addition to changes in neuroinflammation, in which all of these cell types participate. These effects were seen in a sex- and dose-dependent manner.

PFOS exposure has been reported to activate astrocytes in the cortex, resulting in a disrupted blood–brain barrier, and another study has reported that gestational exposure to PFOS results in an increase in GFAP mRNA in the cortex and hippocampus in rats. Together, these studies suggest that astrocyte function is vulnerable to PFOS exposure. In this study, we only probed for gross morphology of astrocytes using IHC, but our transcriptomic data highlight that there may be changes in astrocytes following gestational and lactation exposure to PFHxA in the cerebellum that should be explored using functional assays in the future.

A critical function of OPCs in the developing brain is to mature into myelinating oligodendrocytes. Although myelination does not seem to be affected in our study, it is important to note that our analysis was limited to the granule cell layer and that MBP immunoreactivity allows for only a gross analysis of myelination. Hence, more subtle defects in myelination may be detected using more sensitive techniques such as electron microscopy. Additionally, the myelin that is present may not be functioning the way that it should be for proper axon insulation. Further studies should investigate the function of this myelin for the transduction of electrical potentials.

Immune-related genes were also dysregulated in females. PFAS chemicals have been shown to be immunotoxic as reviewed previously [46,47], including during gestational and lactational exposures. If peripheral macrophages are vulnerable to PFAS exposures [48,49], then it is reasonable to hypothesize that microglia may also be vulnerable to PFAS exposures [50]. At a transcriptional level in females, genes involved in microglia communication with neurons were perturbed, as well as homeostatic genes and genes involved in the complement system. These changes suggest that multiple microglial processes are affected by PFHxA, which would alter the progression of microglia-mediated neurodevelopment and maintenance of the mature brain throughout the organisms’ life. Although there were changes at the transcriptional level, microglia density was not altered in any of the brain regions that we investigated. However, we did observe subtle changes in the morphology of cerebellar microglia, which could indicate changes in microglial function [51,52]. Drawing conclusions based on morphological analysis alone, however, should be considered with caution [53], and future studies should investigate microglia function especially in a lobule-specific manner. One role of microglia during cerebellar development is to phagocytose cleaved caspase-3 expressing PCs [54]. Since we see a PFHxA-induced increase PC linear frequency and perturbations in complement-related genes, it is a possibility that microglia do not properly phagocytose PCs in the higher-treatment group. It should also be considered that subtle changes in PCs and in microglia could together disrupt cellular communication between the two cell types and the changes observed are dependent on each other. Future studies should investigate the interactions of these cell types during neurodevelopment.

### 3.3. Sex Differences

Legacy PFOA and PFOS exposures have been reported to cause sex-specific effects in behavior assays [55,56,57], and we have reported sex-specific effects on adult behavior following developmental PFHxA exposure [17]. However, little is known about the effects of PFAS exposures at the transcriptional level in the brain, although several studies report sex differences in dysregulated gene expression following PFAS exposures in humans in peripheral blood [58] and in the liver of rats [59]. The changes at the transcriptomic level that we report here indicate that the female transcriptome is more vulnerable to PFHxA exposure than the male transcriptome, but this interpretation should be considered with caution. It is also possible that the transcriptomic changes observed in females are transient and do not persist into adulthood where behavioral changes have been observed, while transcriptomic changes may emerge in adulthood in males, causing longer-term alterations in brain function and behavior. Another possibility is that transcriptomic changes occur earlier in males than in females, perhaps at a critical developmental time such as during brain masculinization [60,61,62], leading to changes in circuit formation which could affect brain function in the long-term without being apparent in the bulk-RNA sequencing analysis at P21. The changes could instead be representative of differences in mechanisms of action of PFHxA on the developing female and male, an idea that is supported by the lack of overlap in the genes that were regulated by PFHxA exposure in the two sexes. Understanding the sex-specific timeline of transcriptomic changes would provide a clearer understanding of how and when PFHxA exposure impacts neurodevelopment.

### 3.4. Limitations and Future Directions

The cerebellum is a complex structure with multiple compartments that are recruited in different behavioral circuits. By performing whole tissue bulk-RNA sequencing we could be detecting changes that are reflective of regions outside of the vermis, where we focused our IHC experiments. Future studies should assess phenotypic changes of the cell types investigated here outside of the vermis, and indeed in other brain structures, as well as confirm the differential expression identified via RNA sequencing using other experimental techniques. We also only analyzed a “lower” and “higher” exposure, and this approach does not assess a dose–response relationship that could be elicited by PFHxA, so future studies should utilize multiple doses to understand the dose–response curve elicited by PFHxA, especially since our transcriptomic results suggested non-monotonic effects. Future studies should also consider the potential effects of developmental exposures to PFHxA in mixtures or in comparison to legacy PFAS and address whether the effects we observed in transcriptomic and protein expression persist into adulthood or are resolved.

## 4. Materials and Methods

### 4.1. Animals and Husbandry

C57BL/6J males and females (6 weeks of age) were obtained from the Jackson Laboratory (Bar Harbor, ME, USA) and housed in URMC facilities for 2 weeks under a 12h light/dark cycle at 22 ± 2 °C with chow and water provided ad libitum. Three days before animals were paired, dirty bedding from male cages was placed in female cages in order to synchronize the estrous cycle of the females [63]. Nulliparous females were housed with males and checked daily for the presence of a vaginal plug, and gestational day (GD) 0 was determined once a vaginal plug was observed. On GD0 females were separated from the males and were singly housed for the duration of the study. All experimental protocols were carried out in strict accordance with the University of Rochester Committee on Animal Resources and National Institutes of Health Guidelines.

### 4.2. Developmental Exposure

Offspring were exposed to PFHxA through gestation and lactation by administration of PFHxA through a treat-based method (*Mealworm*, *Tenebrio molitor*) daily to the dam from GD0 until postnatal day (P) 21 (Figure 1A). This administration model reduces maternal stress compared to other oral ingestion route of exposure methods [64]. GD0 females were housed singly once a vaginal plug was observed and were assigned randomly to vehicle control group, 0.32 mg/kg of body weight (bw) PFHxA, or 50 mg/kg of bw PFHxA. PFHxA (CAS: 307-24-4) was purchased from Matrix Scientific (Columbia, SC, USA) and dissolved in double distilled water (ddH_2_O). These two doses were chosen because 0.32 mg/kg of body weight represents the toxicity reference value for humans [15], and the United States Environmental Protection Agency has used a concentration of 50 mg/kg of bw to study the effects of exposure to perfluorohexanoic sulfonic acid (PFHxS), a different 6-carbon chain PFAS, during gestation and lactation [65].

Initially, 20 µL of the solution (concentration of the solution per worm was 0.0004 mg/µL for the 0.32 mg/kg of bw dose, and 0.0625 mg/µL for the 50 mg/kg of bw dose) was injected into the abdomen of each mealworm (Ward’s, supplier VWR, Radnor, PA, USA). This dosing was calculated based on the average weight of a mouse throughout gestation (25 g [66]). To prevent leakage, the mealworm was frozen [64]. The dams were given a PFHxA spiked or vehicle (ddH_2_O) mealworm daily between 9 and 10 am from GD0 until P21 with exception to P0 in order to reduce cage disturbance. The investigator watched the dam consume the entire worm to ensure complete consumption of the assigned dose. On P21, offspring were euthanized, and cerebella were collected for bulk-RNA sequencing or the whole brain was collected for immunohistochemistry. Only one female and one male from an individual litter were used to populate all experiments to eliminate the risk of litter effects. This resulted in six females and six males for each experimental endpoint, which is similar to a sample size that has been used in previous studies investigating developmental insults in cerebellar development [42] and to a study investigating the effects of neurodevelopmental effects of a similar short-chain PFAS [67].

### 4.3. Perfusion and Tissue Collection

Two hours after the final dose of PFHxA was administered, P21 offspring were euthanized with an overdose of sodium pentobarbital and perfused intracardially with 0.1 M phosphate-buffer saline (PBS, pH 7.4).

For RNA sequencing, the whole brain was then harvested, and cerebella were separated from the forebrain. The cerebella were dounce homogenized in 2 mL RLT Plus buffer and 10 uL 2-mercaptoethanol (BME) for RNA extraction. 600 uL of this mixture was flash frozen and subjected to demultiplexing, quality control, alignment, and analysis (described below).

Animals that were used for immunohistochemistry (IHC) were perfused with 4% paraformaldehyde (PFA) directly after PBS perfusion. Brains were extracted and placed in 4% PFA for 24 h at 4 °C then in 30% sucrose at 4 °C until sectioning. Brains were sagittally sectioned on a freezing microtome at 50 µm thickness and stored at −20 °C in cryoprotectant.

### 4.4. RNA Sequencing

Total RNA was isolated using the RNeasy Plus Mini Kit (Qiagen, Valencia, CA, USA) per manufacturers recommendations. The total RNA concentration was determined with the NanopDrop 1000 spectrophotometer (NanoDrop, Wilmington, DE, USA) and RNA quality assessed with the Agilent Bioanalyzer (Agilent, Santa Clara, CA, USA). The TruSeq Stranded mRNA Sample Preparation Kit (Illumina, San Diego, CA, USA) was used for next-generation sequencing library construction per manufacturer’s protocols. Briefly, mRNA was purified from 200 ng total RNA with oligo-dT magnetic beads and fragmented. First-strand cDNA synthesis was performed with random hexamer priming followed by second-strand cDNA synthesis using dUTP incorporation for strand marking. End repair and 3’ adenylation was then performed on the double stranded cDNA. Illumina adaptors were ligated to both ends of the cDNA and amplified with PCR primers specific to the adaptor sequences to generate cDNA amplicons of approximately 200–500 bp in size. The amplified libraries were hybridized to the Illumina flow cell and sequenced using the NovaSeq6000 sequencer (Illumina, San Diego, CA, USA). Single end reads of 100 nt were generated for each sample.

#### Demultiplexing, QC, Alignment, and Analysis

Raw reads generated from the Illumina basecalls were demultiplexed using bcl2fastq version 2.20.0. Quality filtering and adapter removal were performed using FastP version 0.23.1 with the following parameters: “—length_required35—cut_front_window_size 1-cut_front_mean_quality 13—cut_front—cut_tail_window_size 1—cut_tail_mean_quality 13—cut_tail -y –r” [68]. Processed/cleaned reads were then mapped to the GRCm39/gencode M27 reference using STAR_2.7.9a with the following parameters: “—twopass Mode Basic—runMode alignReads—outSAMtype BAM Unsorted—outSAMstrandField intronMotif—outFilterIntronMotifs RemoveNoncanonical–outReadsUnmapped Fastx” [69,70]. Genelevel read quantification was derived using the subread-2.0.1 package (featureCounts) with a GTF annotation file (GRCm39 M27) and the following parameters for stranded RNA libraries “-s 2 -t exon -g gene_name” [71]. Read counts were also quantified using Salmon v1.5.2. The tximport package was used to import and summarize Salmon transcript-level abundance, estimated counts and transcript lengths into gene-level counts and offset matrices for use with downstream gene-level analysis packages. Differential expression analysis was performed using DESeq2-1.34.0 with a logFC 0.25 and *p*-value threshold of 0.05 within R version 3.5.1 (https://www.R-project.org/) [72] and visualized on volcano plots using ggplot2 [73]. A PCA plot was created within R using pcaExplorer to measure sample expression variance [74]. Gene ontology analyses were performed using the EnrichR package with the Biological Processes library logFC 0.25 and *p*-value threshold of 0.05 [75,76,77]. Heatmaps including only significantly differentially expressed genes (*p*-value threshold of 0.05) were generated using the pheatmap package were given the rLog transformed expression values [78]. Heatmaps created for specific cerebellar cell types used cerebellar gene sets that were previously reported in the literature [27], and the genes referenced for heatmaps related to pathways were collected from Mouse Genome Informatics (https://www.informatics.jax.org/). Heatmaps are only shown for females because of the much smaller number of dysregulated genes in males.

### 4.5. Immunohistochemistry

One set of sections was immunoreacted for myelin basic protein (MBP) and calbindin-D-28K. Sections were washed using 0.1 M PBS (pH 7.4) followed by a one hour bovine serum albumin (BSA) block. Sections then underwent overnight incubation at 4 °C (anti-MBP 1:1000, Product: PA1-10008, anti-Calbindin 1:1000, Product: MA5-24135). Next, sections were washed for 10 min 3 times in PBS and incubated for 4 h in secondary antibody solution (Alexa-Fluor 594 (Cat #: A11042) and AlexaFluor 488 (Cat #: A21202), 1:500, Invitrogen, Carlsbad, CA, USA) at room temperature (RT).

A second set of sections were immunoreacted for ionized calcium-binding adaptor molecule 1 (Iba1) and glial fibrillary acidic protein (GFAP). Sections were washed using 0.1 M PBS (pH 7.4) followed by a one-hour bovine serum albumin (BSA) block. Sections then underwent a 2 h RT incubation followed by overnight incubation at 4 °C (anti-Iba1 1:1000, Product: 019-19741 and anti-GFAP 1:1000, Product: AB4674). Next sections were washed for 10 min 3 times in PBS and incubated for 4 h in secondary antibody solution (Alexa-Fluor 594 (Cat #: A21207)) and AlexaFluor 488 (Cat #: A78948), 1:500, Invitrogen, Carlsbad, CA, USA) at RT.

Negative controls sections were included in all IHC experiments. These sections received identical experimental treatment with the absence of incubation in primary antibody solution to confirm the lack of non-specific binding of the secondary antibodies.

### 4.6. Imaging and Analysis

For all imaging experiments, two or three sagittal sections that included the cerebellar vermis or the hippocampus were immunolabeled and imaged. For each animal, the two or three sections of tissue were analyzed and then averaged, and samples were excluded if tissue was torn during experimental processes.

For MBP percent area covered and Purkinje cell linear frequency analysis, images were collected on an Axioplan II (Carl Zeiss, Dublin, CA, USA) epifluorescence microscope with a 10Xobjective (ZEISS Plan-APOCHROMAT, 0.45NA).

For microglia morphology and density analysis and GFAP percent area covered in the cerebellum, images were collected using a Nikon A1R HD confocal microscope using two excitation wavelengths: 488 (filter cube = 450/50) and 561 (filter cube = 525/50) using a 20X (Plan Apo VC 20X DIC N2, MRD70200, 0.75NA) or a 40X water immersion (Apo LWD 40X WI λS DIC N2, MRD77410, 1.15NA) objective.

For microglia density analysis and GFAP percent area covered in the hippocampus, z-stacks of the region of interest were collected with an Olympus BX63 epifluorescence microscope using a 20X (Olympus UPlanFL, 0.50NA) objective.

#### 4.6.1. Purkinje Cell Linear Frequency

Epifluorescence microscope images were collected with a 10x objective and stitched together to examine each lobule of the entire cerebellum to assess Purkinje cell linear frequency. The length of the Purkinje cell layer (PCL) was determined by drawing and measuring a line across the PCL as defined by the presence of Calbindin-positive somas with the segmented line tool in FIJI/ImageJ (1.54 g). The number of PC somas was automatically identified in Cellpose with a custom hand-in-the-loop model based on the cyto2 model. Once an appropriate model was trained in Cellpose, batch processing using this model occurred in Napari to determine the cell counts. The total number of cells for each section was extracted with a custom RStudio (4.3.1) code. To determine the linear frequency, the number of Purkinje cell somas was divided by the length of the PCL for each section [41], and then two or three sections were averaged together to obtain the final value for each animal [42].

#### 4.6.2. Microglia Density

Confocal microscope images were collected using a 40× objective for microglia density analysis in lobule IV/V in the cerebellum, and epifluorescence microscope images were collected with a 20× objective and stitched together for microglia density analysis in the hippocampus. Microglia density was calculated as the number of microglia divided by the area analyzed. Area was determined by drawing and measuring the granule cell layer and the combined Purkinje cell and molecular layer in the cerebellum and the CA1, corpus callosum (CC), and dentate gyrus (DG) in the hippocampus using the polygon tool in FIJI/ImageJ (1.54 g). Microglia were then manually marked using the multipoint tool and counted.

#### 4.6.3. Microglia Morphology: Sholl Analysis

Confocal microscope images were collected with a 40x objective for microglia morphology analysis in lobule IV/V of the cerebellum. Microglia morphology was assessed using Sholl analysis to quantify arbor complexity. In ImageJ/FIJI (1.54 g), 5 individual microglia were selected in the granule cell layer and the molecular layer from maximum projected images (20–25 z-slices). The microglia were outlined and thresholded, and the ImageJ/FIJI (1.54 g) Sholl analysis plug-in was used to create concentric rings at 2 µm intervals out to 60 µm from the soma center. The number of intersections across each ring were calculated to create Sholl curves [79].

#### 4.6.4. Percent Area Covered by MBP and GFAP

The granule cell layer was identified in each lobule of the cerebellum. The background was subtracted and the MBP signal was binarized. The number of signal pixels was measured per area.

Due to the heterogeneity of this astrocytic cell type in the cerebellum, the granule cell layer (GCL) and the combined Purkinje cell and molecular layer (PC and ML) were identified and analyzed separately for each lobule. In the hippocampus, regions of interest were the CA1, CA3, and DG, and the CC was also analyzed using 15 z-slices that were projected for each brain region. The background was subtracted and the GFAP signal was binarized. The number of signal pixels was measured per area [80].

### 4.7. Statistical Analysis

Statistical tests were performed and graphs were generated using GraphPad Prism (10.5.0) IX statistical analysis software (La Jolla, CA, USA). Females and males were analyzed separately in all experiments. For heatmaps, pairwise comparisons to controls were used for the lower- and higher-exposure groups. Immunohistochemistry experiments in all brain regions were analyzed using one-way ANOVAs with Tukey post hoc analyses with exception to the Sholl analyses where Bayesian Hierarchical Sholl models were created and assessed using 95% credible intervals [38]. Detailed statistics can be found in the figure legends.

## 5. Conclusions

In this study, we report the first examination of how PFHxA exposure during gestation and lactation affects the cerebellar transcriptome in a mammalian model. Our results show subtle effects that are most prominent in lower PFHxA exposure in females, suggesting that females are more vulnerable at a late developmental time point. A histological analysis of neurons and glia did not uncover changes in cell numbers in females in the lower-exposure group, although PC linear frequency was increased in lobule III after exposure to the higher dose in both sexes, and alterations in microglial morphology were also observed after exposure to the lower dose in both sexes. This suggests that the subtle transcriptomic changes that occur after developmental PFHxA exposure likely result in alterations of cellular functions rather than grossly affecting cell numbers. Moreover, lower exposure resulted in more changes in the transcriptome, suggesting non-monotonic effects. Together, this suggests that PFHxA at lower or higher doses does not grossly perturb cerebellar development but results in subtle changes to multiple cell types. This finding warrants more study particularly focused on understanding how PFHxA affects cell–cell interactions and the dynamic communication between cell types that is necessary for proper neurodevelopment in a sex-specific manner. This study provides a foundation for understanding the effects of gestational and lactational exposure to PFHxA on the transcriptome of the developing cerebellum, and future studies should investigate the effects on other brain regions such as the hippocampus, striatum, and amygdala.

## Figures and Tables

**Figure 1 ijms-26-08008-f001:**
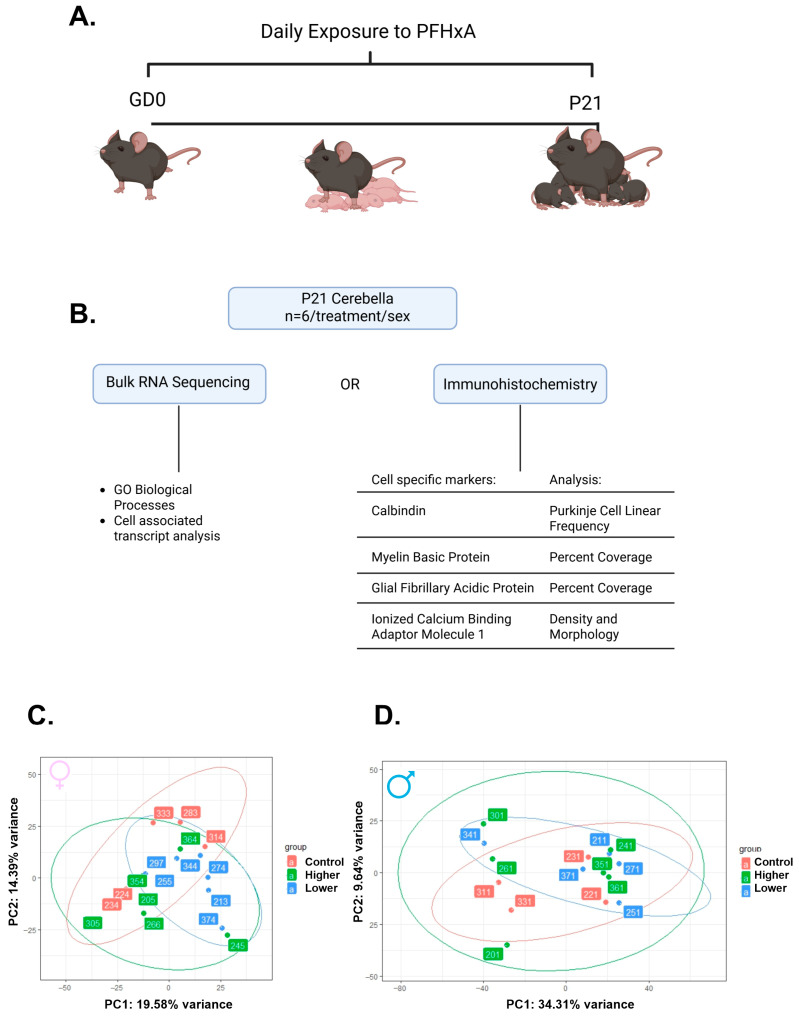
Gestational and lactational exposure to two doses of PFHxA leads to only subtle changes in the cerebellar transcriptome at P21. (**A**) Experimental timeline of daily exposure to PFHxA of one of three doses (ddH20 (control), 0.32 mg/kg of body weight (bw) (lower), or 50 mg/kg of bw (higher)). (**B**) Workflow diagram including the sequencing analysis and immunohistochemistry analyzed. Principal component analysis based on the top 500 variable genes in (**C**) females and (**D**) males (n = 6/treatment/sex).

**Figure 2 ijms-26-08008-f002:**
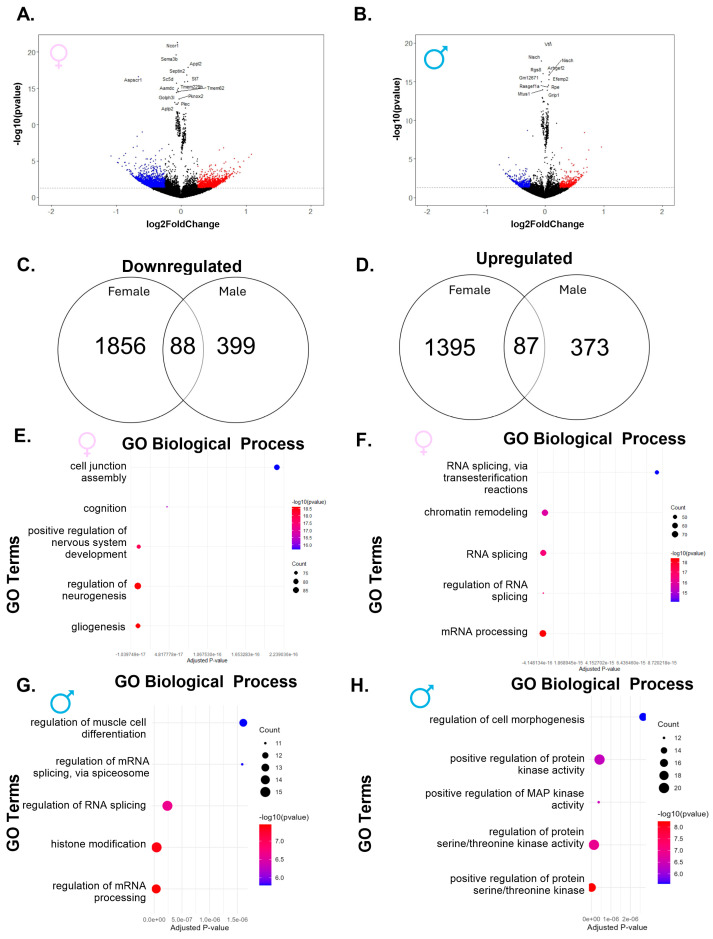
Exposure to the lower dose of PFHxA dysregulates genes in a sex-specific manner when compared to control. Volcano plots in (**A**) females and (**B**) males show significantly upregulated genes in red and significantly downregulated genes in blue. Venn diagrams show more (**C**) downregulated and (**D**) upregulated genes in females as compared to males. Gene ontology (GO) biological processes downregulated (**E**) and upregulated (**F**) in females include more genes than those downregulated (**G**) and upregulated (**H**) in males.

**Figure 3 ijms-26-08008-f003:**
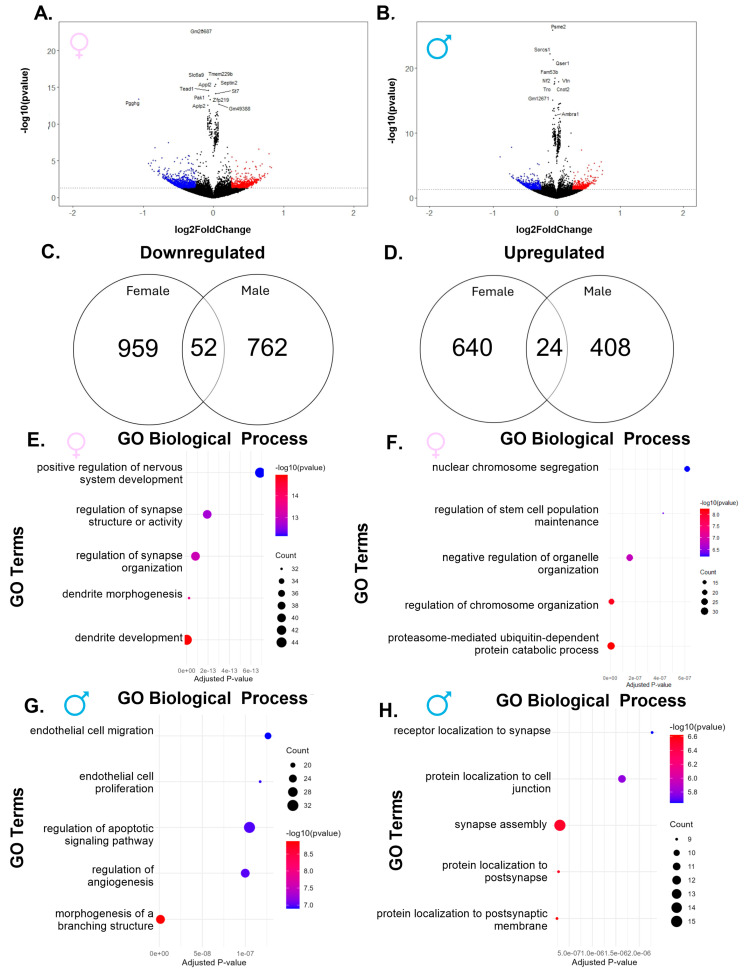
Exposure to the higher dose of PFHxA dysregulates genes in a sex-specific manner when compared to control. Volcano plots in (**A**) females and (**B**) males show significantly upregulated genes in red and significantly downregulated genes in blue. Venn diagrams show more (**C**) downregulated and (**D**) upregulated genes in females as compared to males. In females, there were more genes that belonged to (**E**) downregulated pathways than (**F**) upregulated pathways, while in males, there were a similar number of genes belonging to the (**G**) downregulated and (**H**) upregulated gene ontology (GO) biological processes.

**Figure 4 ijms-26-08008-f004:**
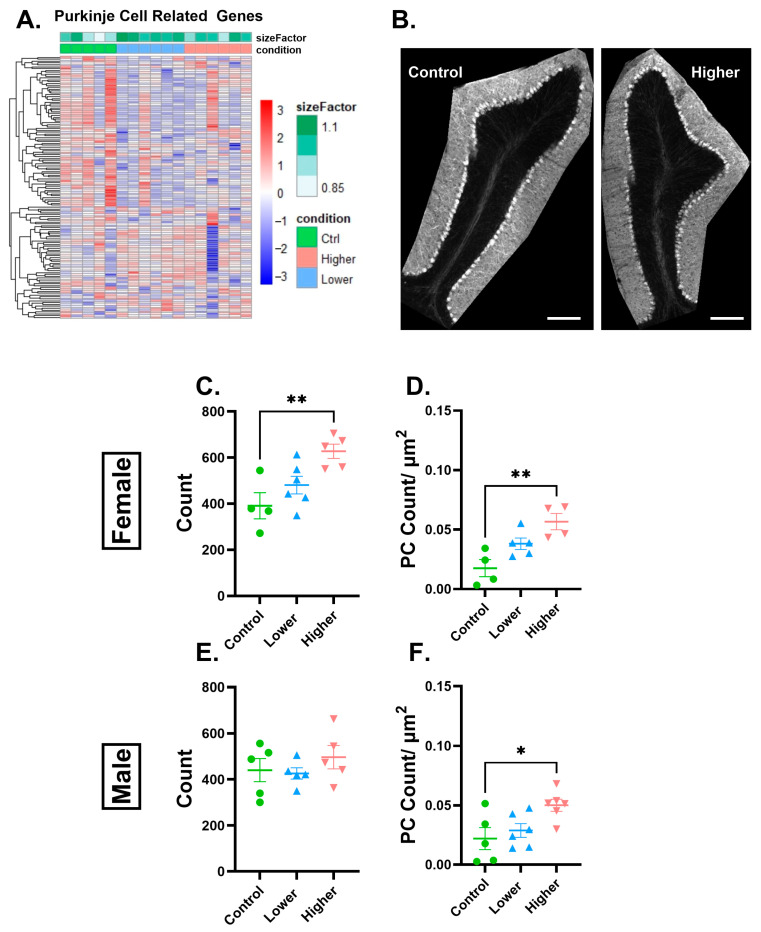
PFHxA exposure affects the Purkinje cell transcriptome and cell count. (**A**) Heatmap of PC-related genes that were dysregulated after lower or higher PFHxA exposure compared to controls in a pairwise comparison in females. (**B**) Representative image of Calbindin1 staining in lobule III. In females, there was a significant increase (**C**) in the PC count in the whole cerebellum (F(2,12) = 7.676, *p* = 0.0071) and (**D**) the PC linear frequency in lobule III between the control and the higher-treatment groups (F(2,10) = 9.329, *p* = 0.0052). In males, there were no changes in the (**E**) PC count in the whole cerebellum, but a significant increase in the (**F**) PC linear frequency between the control and the higher-treatment groups in lobule III (F (2,14) = 4.846, *p* = 0.0252) (**D**). Size factor is the scaling factor to normalize raw read counts for each sample to account for differences in library size. Individual points represent individual animals (n = 4–6). Data are presented as the mean ± SEM. One-way ANOVA with Tukey post hoc analysis (* = *p* < 0.05, ** = *p* < 0.001). Scale bar = 200 µm.

**Figure 5 ijms-26-08008-f005:**
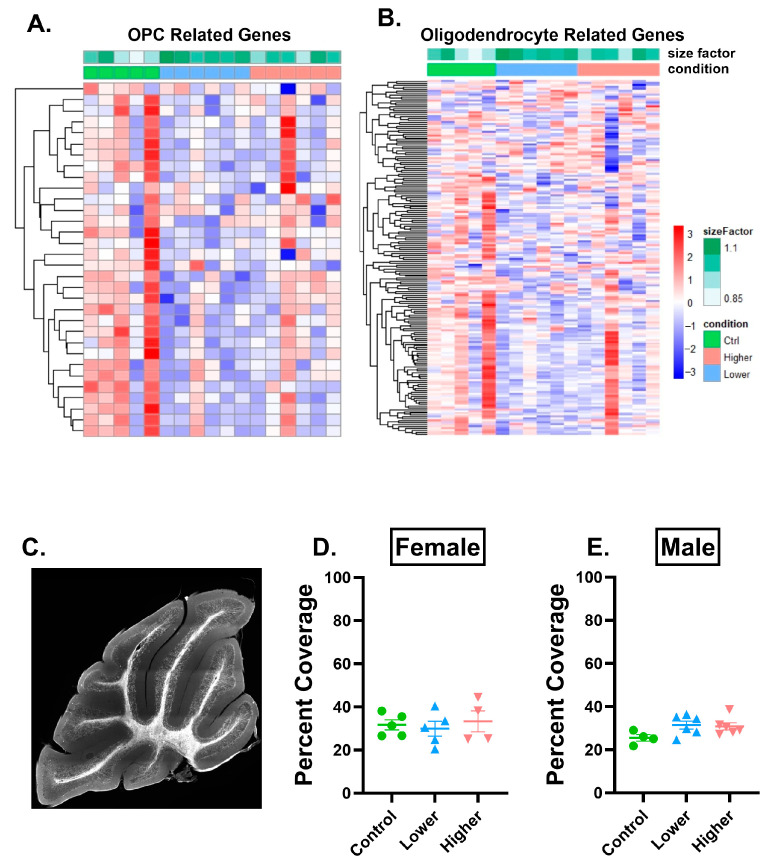
PFHxA exposure affects oligodendrocyte progenitor cell and oligodendrocyte transcripts but does not affect MBP percent area covered in lobule IV/V. (**A**) Heatmap of oligodendrocyte progenitor cell (OPC)-related genes and (**B**) oligodendrocyte-related genes that were dysregulated after lower or higher PFHxA exposure compared to controls using a pairwise comparison in females. (**C**) Representative image of myelin basic protein (MBP) immunolabeling in the whole cerebellum. There were no changes in percent area covered by MBP in (**D**) females or in (**E**) males in lobule IV/V. Size factor is the scaling factor to normalize raw read counts for each sample to account for differences in library size. Individual points represent individual animals (n = 4–6). Data are presented as the mean ± SEM. One-way ANOVA with Tukey post hoc analysis. Scale bar = 500 µm.

**Figure 6 ijms-26-08008-f006:**
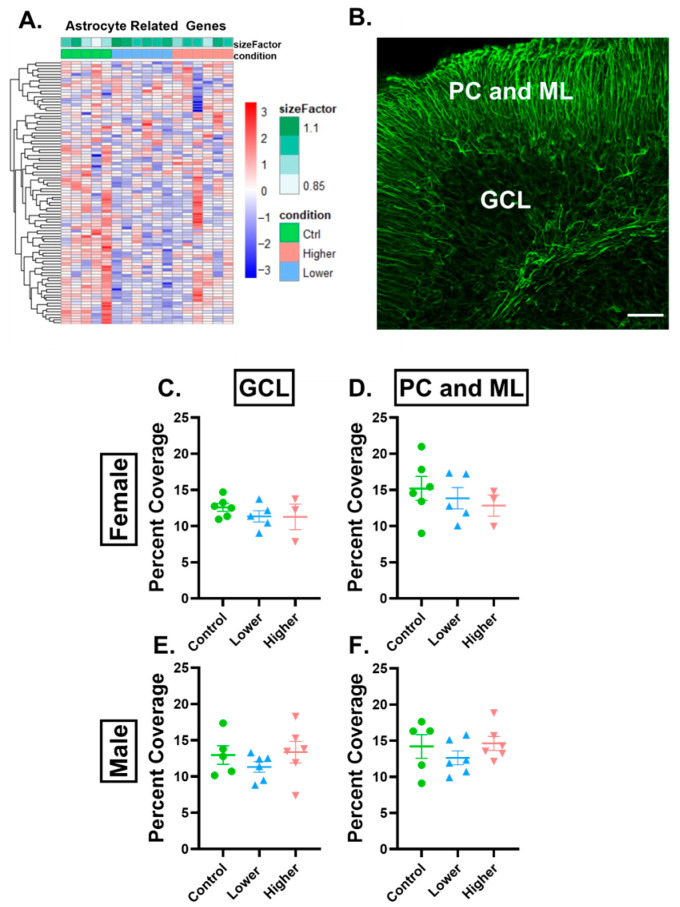
PFHxA exposure affects the astrocyte transcriptome but not GFAP percent coverage in lobule IV/V. (**A**) Heatmap of astrocyte-related genes that were downregulated in the lower- or higher-exposure group compared to controls in a pairwise comparison in females. (**B**) Representative image of GFAP immunolabeling in lobule IV/V of the cerebellum. There were no changes in percent area covered by GFAP in females in the (**C**) granule cell layer (GCL) or the (**D**) Purkinje cell (PC) and molecular layer (ML). In males, there were no changes in percent area covered by GFAP in the (**E**) GCL or the (**F**) PC and ML. Size factor is the scaling factor to normalize raw read counts for each sample to account for differences in library size. Individual points represent individual animals (n = 3–6). Data are presented as the mean ± SEM. One-way ANOVA with Tukey post hoc analysis. Scale bar = 50 µm.

**Figure 7 ijms-26-08008-f007:**
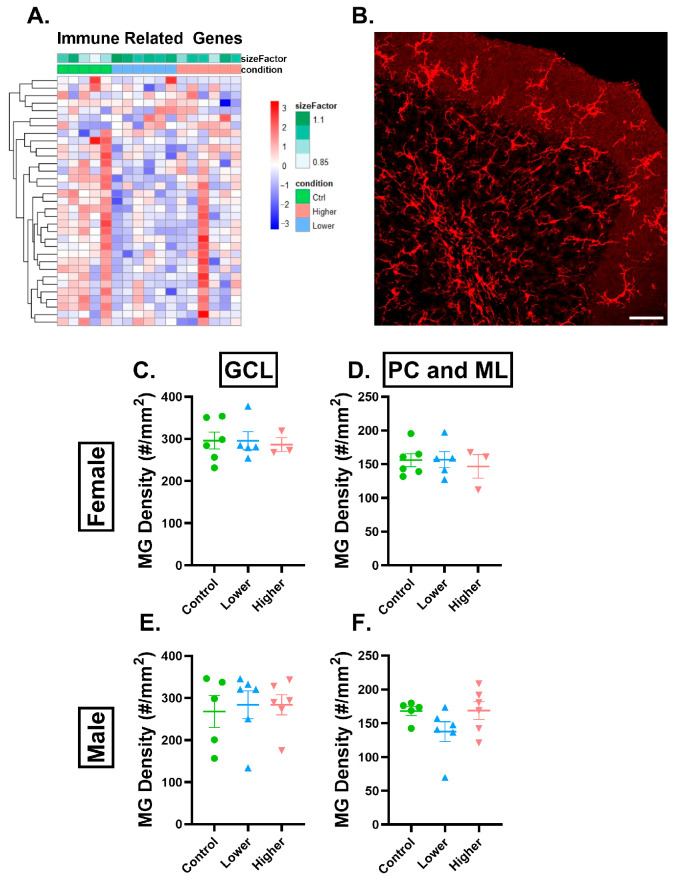
PFHxA exposure affects immune cell transcripts but not microglia density in lobule IV/V. (**A**) Heatmap of immune-related genes that are significantly dysregulated in females from the pairwise comparison in lower or higher exposure compared to controls. (**B**) Representative image of Iba1+ microglia in the granule cell layer (GCL) and Purkinje cell and molecular layer (PC and ML). In females, microglia density is not changed in the (**C**) GCL or in the (**D**) PC and ML. In males, microglia density is not changed in the (**E**) GC layer or in the (**F**) PC and ML. Size factor is the scaling factor to normalize raw read counts for each sample to account for differences in library size. Individual points represent individual animals (n = 3–6). Data are presented as the mean ± SEM. One-way ANOVA with Tukey post hoc analysis. Scale bar = 50 µm.

**Figure 8 ijms-26-08008-f008:**
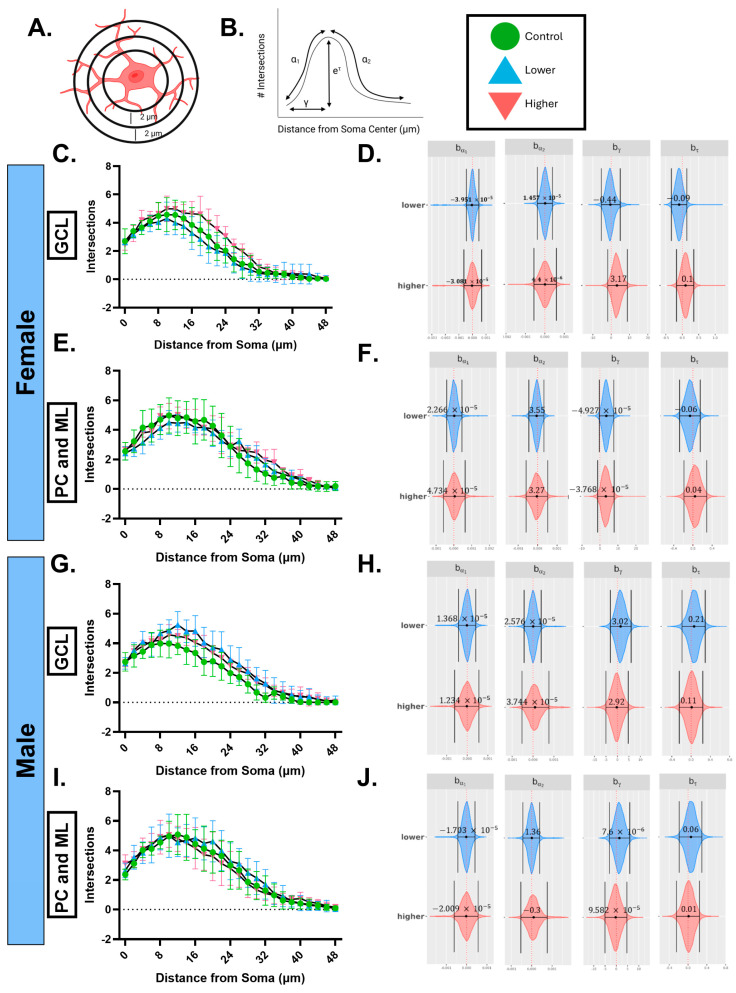
PFHxA exposure subtly affects microglial process ramification in lobule IV/V in females and males. (**A**) Graphical representation of concentric circles around microglia in increasing radii, which corresponds to the *x*-axis of the Sholl curves, and process intersections at the rings correspond to the *y*-axis of the Sholl curves in the granule cell layer (GCL) and Purkinje cell (PC) and molecular layer (ML). (**B**) The Sholl curve is labeled with factors that give information about its behavior (b): before the change point (b_α1_); after the change point (b_α2_); branch maximum (b_eτ_); change point (b_γ_). A hierarchical Bayesian approach was used to fit individual Sholl curves to capture variation at each level of the experimental hierarchy. Credible intervals of 95% for effects of each parameter from (**C**,**E**,**G**,**I**) were calculated to find (**D**,**F**,**H**,**J**), respectively. In the (**C**,**D**) GCL in females, there were no substantial changes, but in the (**E**,**F**) PC and ML there was a substantial shift in the b_γ_ in the lower-exposure group. In the (**G**,**H**) GCL in males, there was a substantial shift in the b_eτ_ in the lower-exposure group, while in the (**I**,**J**) PC and ML, there were no substantial changes. n = 4–6.

## Data Availability

Raw sequencing data were deposited in the NCBI GEO database with accession ID GEO: GSE301375. The RStudio codes and ImgaeJ/FIJI macros can be found at https://github.com/elizacp27/IJMS_Aug2025. The raw imaging data and training models for image analysis will be made available by corresponding author upon reasonable request.

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
