# Peer review of "Gestational and Lactation Exposure to Perfluorohexanoic Acid Results in Sex-Specific Changes in the Cerebellum in Mice"

_ijms, 2025, doi:10.3390/ijms26168008_

Round 1

Reviewer 1 Report

Comments and Suggestions for Authors

General comments:

The manuscript presents an interesting and original study. I have not encountered a similar publication in the existing literature. The methodology is described in a clear and comprehensive manner, which, in my opinion, allows for the reproducibility of the experiments by other researchers. The reference list includes numerous publications from the last five years, which is a significant strength and highlights the relevance of the research topic to the current scientific discourse.

Overall, the results are presented in a clear and accessible manner. However, I suggest the authors consider separating some figure panels or transferring selected content to the Supplementary Materials section, in order to increase the size and clarity of the main figures. This is, of course, left to the authors’ discretion.

Specific comments:

  1. The authors present a wide range of analyses. To better structure the flow of the study, I recommend including a workflow diagram at the beginning of the Results and Discussion
  2. Figure 6B is rather difficult to interpret due to its density. It may be worth considering dividing it into separate figures to improve readability and increase the size of each panel.
  3. On page 8, the section on membrane potential regulation should be expanded. I suggest providing a more detailed discussion of the mechanism, ideally supported by relevant literature.
  4. In Figure 9, the standard deviations appear to be relatively large. Please explain the potential reasons for this variability. Is it typical for this type of measurement, or could it be specific to this study?
  5. The Conclusions section is interesting, however I would encourage the authors to elaborate further on the practical implications of the study, critically address its limitations, and outline perspectives for future research.

Author Response

Reviewer 1

General comments:

The manuscript presents an interesting and original study. I have not encountered a similar publication in the existing literature. The methodology is described in a clear and comprehensive manner, which, in my opinion, allows for the reproducibility of the experiments by other researchers. The reference list includes numerous publications from the last five years, which is a significant strength and highlights the relevance of the research topic to the current scientific discourse.

Overall, the results are presented in a clear and accessible manner. However, I suggest the authors consider separating some figure panels or transferring selected content to the Supplementary Materials section, in order to increase the size and clarity of the main figures. This is, of course, left to the authors’ discretion.

Specific comments:

The authors present a wide range of analyses.

  1. To better structure the flow of the study, I recommend including a workflow diagram at the beginning of the Results and Discussion

We appreciate this reviewer’s suggestion, and we agree that this would better structure the flow of the study. We have included a workflow diagram in Figure 1.

  1. Figure 6B is rather difficult to interpret due to its density. It may be worth considering dividing it into separate figures to improve readability and increase the size of each panel.

We agree that the readability of this figure could be improved. Therefore, we have rearranged the panels to extend the size of the heatmaps.

  1. On page 8, the section on membrane potential regulation should be expanded. I suggest providing a more detailed discussion of the mechanism, ideally supported by relevant literature.

Considering feedback from Reviewer 2 we have removed the pathways analyses that were originally in Figure 4 and moved the volcano plots and Venn Diagrams to supplementary Figure 1, eliminating this discussion from the manuscript.

  1. In Figure 9, the standard deviations appear to be relatively large. Please explain the potential reasons for this variability. Is it typical for this type of measurement, or could it be specific to this study?

We agree that the standard deviations appear relatively large, and this is consistent for Sholl studies across the literature. Given the heterogeneity of microglial populations, microglial biologists are still trying to determine the most reliable method  to analyze microglia morphology (https://pubmed.ncbi.nlm.nih.gov/37637472/), which is why we also included a model-based hierarchical Bayesian approach (https://academic.oup.com/bioinformatics/article/40/4/btae156/7633407).

  1. The Conclusions section is interesting, however I would encourage the authors to elaborate further on the practical implications of the study, critically address its limitations, and outline perspectives for future research.

In light of reviewer 2’s comments we did not want to overly expand our discussion and conclusions sections. However, we appreciate the reviewer’s comment and we have expanded on the discussion section of the limitations for the study (see section 3.4) and we added the following perspective for future research in the conclusion (section 5): “This study provides a foundation for understanding the effects of gestational and lactational exposure to PFHxA on the transcriptome of the developing cerebellum, and future studies should investigate the effects on other brain regions such as the hippocampus, striatum, and amygdala.”

Reviewer 2 Report

Comments and Suggestions for Authors

The manuscript covers an interesting topic, but it is very long, and it hampered by a major limitation, i.e. too many data are described which are in Supplementary figures (7, and with multiple panels), but the reference to Supplementary material should be limited, otherwise the manuscript cannot be fully understood and fully clear on the main findings. The description of Figure 1, 2 and 3 is too long and one of these figures should be eliminated, avoiding the repetitive comparison of low doses and higher doses of treatment (Figures 1 and 2).

Major points

In the Abstract, the definition of “legacy PFAS” should be introduced to readers

In the Abstract, the sentence “Given the high PFHxA levels in cerebellum and protracted developmental window, acute transcriptional dysregulation and cellular morphology of the cerebellum was assessed ...” is unclear, as it is difficult to catch if the high PFHxA levels in cerebellum were measured in the manuscript or based on the previous studies

In the Results section, main text, it is difficult to understand how many pups, either female or male, were used

The final sentence of Section 2 Results should be the Introduction of Section 2.1, which otherwise it is difficult to catch. In addition, Section 2.1 begins with data that are in the Supplementary material, but it should begin with data that are shown, such as those in Figure 5

In Section 2.1, please briefly illustrate the biochemical function of dysregulated genes, such as Lrrc24 and Ptgfrn.

In Section 2.1, female gene sets were used, but to address changes in PC cell numbers female and male samples were used. This generates confusion

The comment to Supplementary Figure 3 is unclear and unnecessary

The final part of Section 2.1 and of section 2.2 on glial cells, astrocytes and microglia all end up with the description of data reported in Supplementary figures, and this should be avoided or be very synthetic.

The Discussion section is also too long and should compacted, to avoid repetitive and too general introduction on neuronal cells and focusing the manuscript main findings.

In 3.4. Limitations and Future Directions, authors should mention that no differential expression of genes was confirmed, as for example by in situ hydridization.

Author Response

Reviewer 2

Comments and Suggestions for Authors
The manuscript covers an interesting topic, but it is very long, and it hampered by a major limitation, i.e. too many data are described which are in Supplementary figures (7, and with multiple panels), but the reference to Supplementary material should be limited, otherwise the manuscript cannot be fully understood and fully clear on the main findings.

  1. The description of Figure 1, 2 and 3 is too long and one of these figures should be eliminated, avoiding the repetitive comparison of low doses and higher doses of treatment (Figures 1 and 2).

We believe that the information contained in Figures 1-3 is critical to the manuscript. However, we appreciate this feedback, and we have eliminated the pathway analysis of figure 4 and moved the volcano plots and Venn diagrams to supplementary Figure 1 to limit the repetitive comparison of the effects of different doses.

  1. In the Abstract, the definition of “legacy PFAS” should be introduced to readers

We appreciate this suggestion, and we exchanged “legacy” with “currently regulated” which more clearly describes the PFAS to which we are referring.

  1. In the Abstract, the sentence “Given the high PFHxA levels in cerebellum and protracted developmental window, acute transcriptional dysregulation and cellular morphology of the cerebellum was assessed ...” is unclear, as it is difficult to catch if the high PFHxA levels in cerebellum were measured in the manuscript or based on the previous studies

We appreciate this observation, and we agree that the way it was worded was unclear. To provide clarity we have changed the sentence to: “Given the high PFHxA levels in the cerebellum in post-mortem studies and the cerebellum’s protracted developmental window, we assessed acute transcriptional dysregulation and cellular morphology of this brain region on the last day of exposure at P21.”

  1. In the Results section, main text, it is difficult to understand how many pups, either female or male, were used

We agree that it is unclear how many pups were used, therefore we added this information to the workflow panel in Figure 1 as well as the legend “(n=6/treatment/sex)”.

  1. The final sentence of Section 2 Results should be the Introduction of Section 2.1, which otherwise it is difficult to catch. In addition, Section 2.1 begins with data that are in the Supplementary material, but it should begin with data that are shown, such as those in Figure 5

We agree with this reviewer that the results from Figure 5 should lead this section, so we have reordered this section to make it less confusing:  “We first address Purkinje cells (PCs), the main output neuron of the cerebellum. 134 genes in lower exposure, 26 in higher exposure, as well as 20 genes that were in common to both exposures were dysregulated in a gene set that represented PC related genes (Figure 5) 27. Along with PC related genes, genes involved in both neurotransmitter transport and axon guidance were dysregulated, suggesting that cerebellar neurons may be sensitive to PFHxA exposure (Supplementary Figure 2).”

  1. In Section 2.1, please briefly illustrate the biochemical function of dysregulated genes, such as Lrrc24 and Ptgfrn.

We agree that this is important information to include. Now the sentence reads: “For example, Lrrc24 and Ptgfrn are both expressed throughout the lifetime by PCs but their biochemical function is unknown, while Grfa2 and Itga3 are involved in PC development 28,29 and are downregulated and upregulated respectively, and Cttnbp2 is a synaptic gene that is downregulated and has been implicated in neurodevelopmental disorders 30,31.”

  1. In Section 2.1, female gene sets were used, but to address changes in PC cell numbers female and male samples were used. This generates confusion

We appreciate that this could cause confusion. In 4.4.1. we explain that because of the much smaller number of dysregulated genes in males, we only show the heatmaps for females. We consistently found much fewer dysregulated genes for specific cell types in males. Because of this, we excluded that data from this paper.

  1. The comment to Supplementary Figure 3 is unclear and unnecessary

We agree that it is unclear, so we took out the comment to Supplementary Figure 3 and changed the previous sentence to: “Interestingly, in both males and females we found a significant increase in PC linear frequency in the higher treatment group in only lobule III (Figure 5D and 5F, Supplementary Figure 3).”

  1. The final part of Section 2.1 and of section 2.2 on glial cells, astrocytes and microglia all end up with the description of data reported in Supplementary figures, and this should be avoided or be very synthetic.

We agree that we included a lot of description of the supplementary figures, so we have combined and condensed that information in the respective paragraphs.

  1. The Discussion section is also too long and should compacted, to avoid repetitive and too general introduction on neuronal cells and focusing the manuscript main findings.

We have compacted the discussion by removing information that had been previously discussed in the manuscript while keeping the main findings.

  1. In 3.4. Limitations and Future Directions, authors should mention that no differential expression of genes was confirmed, as for example by in situ hydridization.

We agree with the reviewer that this is a very important point. We added this point in 3.4. “Future studies should assess phenotypic changes of the cell types investigated here outside of the vermis, and indeed in other brain structures, as well as confirm the differential expression identified via RNA sequencing using other experimental techniques.”

Round 2

Reviewer 2 Report

Comments and Suggestions for Authors

Authors satisfactorily addressed all raised issues